# Dual PI3K/mTOR Inhibitor NVP-BEZ235 Leads to a Synergistic Enhancement of Cisplatin and Radiation in Both HPV-Negative and -Positive HNSCC Cell Lines

**DOI:** 10.3390/cancers14133160

**Published:** 2022-06-28

**Authors:** Florentine S. B. Subtil, Carolin Gröbner, Niklas Recknagel, Ann Christin Parplys, Sibylla Kohl, Andrea Arenz, Fabian Eberle, Ekkehard Dikomey, Rita Engenhart-Cabillic, Ulrike Schötz

**Affiliations:** 1Department of Radiotherapy and Radiooncology, Philipps-University, 35043 Marburg, Germany; florentine.subtil@staff.uni-marburg.de (F.S.B.S.); groeca@web.de (C.G.); recknage@students.uni-marburg.de (N.R.); parplys@staff.uni-marburg.de (A.C.P.); kohlsi@students.uni-marburg.de (S.K.); arenza@staff.uni-marburg.de (A.A.); fabian.eberle@uk-gm.de (F.E.); dikomey@uke.de (E.D.); rita.engenhart-cabillic@uk-gm.de (R.E.-C.); 2Laboratory of Radiobiology & Experimental Radiooncology, University Medical Center Hamburg-Eppendorf, 20251 Hamburg, Germany

**Keywords:** HNSCC, HPV, NVP-BEZ235, irradiation, cisplatin, clonogenic survival, radiosensitization

## Abstract

**Simple Summary:**

Head and neck cancers (HNSCCs), especially in the advanced stages, are predominantly treated by radiochemotherapy, including cisplatin. The cure rates are clearly higher for HPV-positive HNSCCs when compared to HPV-negative HNSCCs. For both entities, this treatment is accompanied by serious adverse reactions, mainly due to cisplatin administration. We reported earlier that for both HPV-positive and negative HNSCC cells, the effect of radiotherapy was strongly enhanced when pretreated using the dual PI3K/mTOR inhibitor NVP-BEZ235 (BEZ235). The current study shows that for HPV-positive cells, BEZ235 will strongly enhance the effect of cisplatin alone. More important, preincubation with BEZ235 was found to alter the purely additive effect normally seen when cisplatin is combined with radiation into a strong synergistic enhancement. This tri-modal combination might allow for the enhancement of the effect of radiochemotherapy, even with reduced cisplatin.

**Abstract:**

The standard of care for advanced head and neck cancers (HNSCCs) is radiochemotherapy, including cisplatin. This treatment results in a cure rate of approximately 85% for oropharyngeal HPV-positive HNSCCs, in contrast to only 50% for HPV-negative HNSCCs, and is accompanied by severe side effects for both entities. Therefore, innovative treatment modalities are required, resulting in a better outcome for HPV-negative HNSCCs, and lowering the adverse effects for both entities. The effect of the dual PI3K/mTOR inhibitor NVP-BEZ235 on a combined treatment with cisplatin and radiation was studied in six HPV-negative and six HPV-positive HNSCC cell lines. Cisplatin alone was slightly more effective in HPV-positive cells. This could be attributed to a defect in homologous recombination, as demonstrated by depleting RAD51. Solely for HPV-positive cells, pretreatment with BEZ235 resulted in enhanced cisplatin sensitivity. For the combination of cisplatin and radiation, additive effects were observed. However, when pretreated with BEZ235, this combination changed into a synergistic interaction, with a slightly stronger enhancement for HPV-positive cells. This increase could be attributed to a diminished degree of DSB repair in G1, as visualized via the detection of γH2AX/53BP1 foci. BEZ235 can be used to enhance the effect of combined treatment with cisplatin and radiation in both HPV-negative and -positive HNSCCs.

## 1. Introduction

Advanced HNSCCs are mainly treated with radiotherapy, combined with concurrent cisplatin-based chemotherapy [1,2]. For HPV-positive (HPV pos.) oropharyngeal HNSCCs, this therapy modality leads to a 5-year-survival rate of approximately 85%, in contrast to only 50% for HPV-negative (HPV neg.) HNSCCs [3,4].

The combination of cisplatin and radiation was established at the beginning of this century by several large trials showing that the overall survival of HNSCC patients was enhanced by 8 to 11% when compared to radiation alone [5,6,7]. Further enhancement was achieved by radiochemotherapy (RCT) when cisplatin was combined with docetaxel and 5-Fluorouracil [8].

Cisplatin is a planar molecule that needs to be activated within the cell, which is then able to bind to the DNA, causing a variety of DNA crosslinks leading either to intra-strand (90%) or inter-strand crosslinks (ICLs, 3–5%) [9]. These lesions are repaired by using almost all major DNA repair pathways, including nucleotide excision repair (NER), homologous recombination (HR), and non-homologous end-joining (NHEJ) [9,10,11,12]. While intra-strand crosslinks are primarily processed by NER, inter-strand crosslinks are mostly repaired by NHEJ and HR. Cisplatin is considered to be especially effective in S phase cells, when crosslinks may interfere with replication, and therefore, cisplatin is seen as an optimal tool to kill proliferating tumor cells, which are generally more radioresistant [13,14].

However, it was noted that RCT with cisplatin is accompanied by severe side effects [5,6]. Therefore, there has always been great interest in establishing new treatment modalities, which would result in a better outcome for HPV neg. HNSCC, but most of all, in less side effects. Unfortunately, clinical studies with EGFR inhibitors combined with RCT failed for HPV neg. HNSCC [15,16,17,18], and also, for patients with HPV pos. HNSCC, treatment with EGFR inhibitors in combination with radiotherapy alone did not result in a better outcome when compared to RCT with cisplatin [19,20,21]. Therefore, cisplatin-based RCT is still considered to be the standard of care for both HPV neg. and pos. HNSCC [22].

It is generally assumed that therapy might be improved for HNSCC when using molecular targeting. In this respect, the PI3K/Akt/mTOR pathway is regarded as being an optimal target, since it is often mutated in HPV pos. and HPV neg. HNSCCs [23,24,25,26,27], and the mutations are correlated with a bad prognosis, and resistance to both radio- and chemotherapy [28,29].

The dual inhibitor NVP-BEZ235 (BEZ235, Dactolisib) can obtain a strong suppression of the dual PI3K/mTOR pathway [30]. When combined with radiation, BEZ235 was also found to result in strongly enhanced radiosensitivity [31,32,33,34,35,36]. In contrast, for other PI3K-inhibitors such as BKM120 and GDC0980, no such enhancement was obtained [37].

It was recently reported by us [38] that BEZ235 can also be used to increase the radiosensitivity of both HPV neg. and pos. HNSCC cells. This effect was not due to enhanced apoptosis, but resulted from suppressed double strand break (DSB) repair by non-homologous end-joining (NHEJ), which is the DSB pathway that is mainly active in G1 phase cells [39]. In contrast, no effect was seen for DSB repair by homologous recombination (HR), which was previously found to be defective in HPV pos. HNSCC cells [40].

It was tested for HPV neg. and pos. HNSCC cells whether BEZ235 could also be used to enhance the effect of cisplatin alone, but most of all, the effect of the combined treatment with cisplatin and radiation. These experiments were performed with six HPV neg., as well as six HPV pos. HNSCC cell lines, to cover the huge molecular heterogeneity generally seen for this entity [41]. It was found that BEZ235 can be used to enhance the effect of cisplatin alone, but solely for HPV pos. cells, and that this is due to their defective HR. Most notably, BEZ235 was found to enhance the combined effects of cisplatin and radiation in both, HPV neg. and pos. HNSCC cells, thereby converting the mostly additive effect seen for this combination into a synergistic interaction.

## 2. Materials and Methods

### 2.1. Cell Lines and Culture Conditions

Experiments were performed with six HPV neg. (FaDu, UM-SCC-3, UM-SCC-6, UM-SCC-11b, UT-SCC-33, and Cal-33) and six HPV pos. (UD-SCC-2, UM-SCC-47, UM-SCC-104, 93VU-147T, UPCI:SCC-152, and UPCI:SCC-154) HNSCC cell lines, and OKF6 a normal human oral keratinocyte. The authentication of all cell lines was verified using Short Tandem Repeats (STR) analysis at the Helmholtz Center Munich using the GenePrint 10 kit (Promega, Mannheim, Germany) and GeneMapper 5.0 software. The resulting data were compared with the Expasy and DSMZ databases [42]. HNSCC cells were maintained in RPMI-1640 medium (Sigma Aldrich, Munich, Germany) or DMEM (Life Technologies, Darmstadt, Germany) supplemented with 10% FBS (Biochrom, Berlin, Germany), 2 mM L-glutamine (Capricorn Scientific GmbH, Ebsdorfergrund, Germany), and 1% non-essential amino acids (PAA Laboratories GmbH, Pasching, Austria). OKF6 cells were cultivated in a 2:1:1 mix of keratinocyte SFM:DMEM:F12 medium containing 40 µg/mL BPE, 0.03 ng/mL hurEGF, 25 U/mL penicillin, 25 µg/mL streptomycin, and 155 µM CaCl2 (all from Life Technologies, Darmstadt, Germany). Cultures were maintained at 37 °C in a humidified 5% CO_2_ atmosphere and routinely tested for mycoplasma contamination [43].

### 2.2. BEZ235 and Cisplatin Treatment and Radiation

If not stated otherwise, NVP-BEZ235 (Selleckchem, Munich, Germany; [44]) was applied in a final concentration of 50 nM in DMSO (Dimethylsulfoxid; AppliChem GmbH, Darmstadt, Germany) 2 h prior to cisplatin treatment and/or irradiation, and removed 26 h later by a medium change.

Stock solutions of cisplatin (0.33 mg/mL, Teva GmbH, Ulm, Germany) were prepared by the Center for Cytostatics Preparation, University Hospital Giessen and Marburg, Marburg, Germany, and diluted in medium to generate indicated concentrations. In the case of combined treatments, cisplatin was added to the medium before irradiation. Cisplatin was removed by a medium change 24 h afterwards.

Cells were irradiated with X-rays using a Precision X-RAD 320ix (Precision X-ray, North Branford, CT, USA) at 320 kV and 8 mA, a dose rate of 1.1 Gy/min, and a Thoräus filter of 0.5 mm Cu + 0.5 mm Al. Absolute dose measurements confirmed the applied doses.

### 2.3. Specific Targeting of RAD51 by siRNA Transfection

Transfection was carried out as described previously [45] using Lipofectamine 2000 transfection reagent (Life Technologies, Carlsbad, CA, USA) according to the manufacturer’s instructions, with RAD51-specific siRNA oligonucleotides (J-003530-09: UAUCAUCGCCCAUGCAUCA, J-003530-10: CUAAUCAGGUGGUAGCUCA, J-003530-11: GCAGUGAUGUCCUGGAUAA, J-003530-12: CCAACGAUGUGAAGAAAUU) or non-targeting siRNA (D-001810-01: UGGUUUACAUGUCGACUAA, D-001810-02: UGGUUUACAUGUUUUCUGA, D-001810-03: UGGUUUACAUGUUGUGUGA, D-001810-04: UGGUUUACAUGUUUUCCUA) purchased from Dharmacon (ON-Targetplus, SMARTpool; Horizon Discovery Group, Cambridge, UK). A total of 1–1.5 × 10^5^ cells were seeded to the wells of six-well plates and left overnight for adherence. For transfection, cells were incubated 4 h with a mix of 20–50 nM siRNA and 2–5 µL lipofectamine 2000 in OPTI-MEM (Life Technologies, Carlsbad, CA, USA). Cells were used for further experiments 1 day after transfection.

### 2.4. Colony Formation Assay

Cells were seeded in triplicate in 6-well plates in various cell numbers, depending on the cell line. In the case of BEZ235, cells were treated 2 h before cisplatin and/or irradiation. All inhibitors were washed out 24 h after irradiation by a medium change. Cells were left to grow until the colonies of the treatment arm had reached equal colony size to the control arm (approximately 10–28 days), and then they were then fixed and stained for colony counting (≥50 cells). Clonogenicity was analyzed by colony formation assay, as described previously [46]. The surviving fractions (SF) of the samples were normalized to the plating efficiency of the respective control.

### 2.5. Western Blot Analysis

Following treatment, cells were lysed in ice cold RIPA buffer supplemented with protease inhibitor cocktail and PMSF (AppliChem, Darmstadt, Germany). Lysates were boiled in 1× SDS-PAGE sample buffer (25 mM Tris-HCl, pH 6.8; 10% glycerol, 2% SDS, 2.5% β-mercaptoethanol, and 0.005% bromophenol blue) and equivalent amounts of protein were electrophoresed on SDS-PAGE gels. PVDF membranes were blotted with antibodies that recognize Rad51 (#ab133534, 1:10,000, Abcam, Cambridge, UK), β-actin (clone AC-15, #A5441, 1:5000, Sigma-Aldrich, St. Louis, MI, USA), and α-tubulin (#sc-5286, 1:2000, Santa Cruz Biotechnology, Inc., Santa Cruz, CA, USA), and subsequently incubated with anti-rabbit/anti-mouse IgG HRP (horseradish peroxidase)-linked antibodies (1:5000, Millipore, Darmstadt, Germany). For the cell line UD-SCC-2, β-actin was used as loading control, because this cell line did not express α-tubulin. Proteins were visualized using an ECL chemiluminescence detection system (Amersham, Munich, Germany).

### 2.6. Immunofluorescent Microscopy

Immunofluorescence analysis was performed as previously described [47]. Briefly, cells were grown on glass coverslips 24 h before treatment, and 24 h later, cells were washed with PBS and fixed with 4% paraformaldehyde for 10 min. Fixed cells were permeabilized with 0.2% Triton X-100 on ice for 5 min. The cells were incubated with primary antibodies: anti-γ-H2AX (Millipore, Darmstadt, Germany), and permeabilized with 0.1% Triton X-100, 1.5% BSA/PBS, for 10 min, and blocked in 3% BSA/PBS at 4 °C overnight. Primary antibody incubation was performed for 1 h at room temperature using the following antibodies: mouse monoclonal anti-phospho-S139-H2AX antibody (1:500, clone: JBW301, Millipore, Darmstadt, Germany) and rabbit polyclonal anti-53BP1 antibody (1:500, #NB100-305, Novus Biologicals, Centennial, CO, USA). After washing three times with 0.5% Tween20/PBS for 10 min, the cells were incubated for 1 h with secondary goat-anti-mouse Alexa-Fluor647 IgG (1:1000, Invitrogen, Waltham, MA, USA) and donkey-anti-rabbit Alexa-Fluor488 (1:1000, Invitrogen). Cells were again washed three times and mounted in ProLong Gold antifade reagent (Invitrogen, Karlsruhe, Germany), including DAPI, for the staining of nuclei. Immunofluorescence was analyzed using the DM5500 B microscope (objective: 63x, Leica, Wetzlar, Germany) and LAS AF and LAS X Software (Leica, Wetzlar, Germany). For analysis, z-stacked images were taken from each sample and the foci were counted manually. The number of foci in irradiated samples was calculated by background subtraction from non-irradiated controls. All experiments were performed at least twice in duplicates, and at least 100 nuclei were counted, which is in line with actual recommendations [https://radbiolab.shinyapps.io/terrific; accessed on 21 June 2022].

### 2.7. Statistical Analysis

If not stated otherwise, all experiments were performed in triplicates and repeated at least three times. Data are presented as the mean values ± standard error of the mean (SEM). A Student’s *t*-test was used to check for statistical significance, with a significance level *p* < 0.05. All statistical analysis and graphics were made using GraphPad Prism version 9.0 (GraphPad Software Inc., La Jolla, CA, USA).

## 3. Results

### 3.1. Cisplatin Sensitivity Is Slightly Higher for HPV Pos. HNSCC Cell Lines

Figure 1 shows the effect of cisplatin on HPV neg. and pos. HNSCC cell lines. Cells were incubated for 24 h, with cisplatin concentrations of up to 2.0 µM, followed by a colony formation assay (Figure 1). As shown for one HPV neg. and one HPV pos. cell line, survival decreases with an increasing concentration of cisplatin (Figure 1a,b). These data sets were used to determine the half-maximal inhibitory concentration (IC50), and accordingly, for the other five HPV pos. and five HPV neg. HNSCC cell lines, as well (Appendix A; Figure 1c). Except for the UD-SCC-2 cells, HPV pos. cell lines were, on average, slightly more sensitive than HPV neg. cell lines, with respective mean values of 0.17 ± 0.06 µM and 0.35 ± 0.10 µM, and with a *p* value of *p* = 0.119 and *p* = 0.037 when UD-SCC-2 was excluded.

### 3.2. Knockdown of RAD51 Increases Cisplatin Sensitivity of HPV Neg. but Not of HPV Pos. Cells

HPV pos. cells are defective in DSB repair performed by HR, as demonstrated by the lack of formation of RAD51 foci after X-irradiation [40]. In order to test whether this defect might also explain the different degrees of cisplatin sensitivity described above, RAD51 was knocked-down (KD) in both HPV neg. and pos. cell lines using specific siRNA (Figure 2a, Appendix A). This experiment was especially performed with cell lines showing high IC50 values. There was no difference in the basal expression of RAD51 between these two groups (Appendix A). For the HPV neg. HNSCC cell line Cal-33, the KD was found to enhance cisplatin sensitivity significantly. This, however, was not seen for the HPV pos. cell line, UM-SCC-47 (Figure 2b,c). Similar results were obtained for two other HPV neg. and one pos. cell lines (Appendix A). The finding that KD of RAD51 led to a reduced IC50 solely for HPV neg. cells but not for HPV pos. cell lines (Figure 2d) demonstrated that the higher cisplatin sensitivity of the HPV pos. cells seems to be linked to their defect in HR.

### 3.3. BEZ235 Enhances Cisplatin Sensitivity of HPV Pos. but Not of HPV Neg. Cells

It was recently shown by us that a pretreatment with BEZ235, which is a potent dual inhibitor of the PI3K/mTOR signaling pathway, resulted in an enhanced degree of radiosensitivity for both HPV neg. and pos. HNSCC cells, and that this was due to an inhibited NHEJ [38]. It was now tested whether BEZ235 might also have an impact on cisplatin sensitivity. BEZ235 was given 2 h prior to cisplatin, followed by further incubation for 24 h, before the medium was changed for the colony formation assay (Figure 3). The time interval of 2 h prior to cisplatin treatment was chosen because of the strongest effect measured for this combination (Appendix A). BEZ alone was found to have almost no effect on the survival of HPV neg. cells, and only a very modest effect on HPV pos. cells, with the only exception of UM-SCC-47 cells (Figure 3a). For the HPV neg. cell line, Cal-33 pretreatment with BEZ235 was found to have no effect on cisplatin sensitivity, in contrast to the clear enhancement seen for the HPV pos. UM-SCC-47 cells (Figure 3b). In Figure 3c, the effect of BEZ235 on cisplatin sensitivity is presented for all 12 HPV neg. and pos. cell lines, whereby for cisplatin, IC50 was always used, as taken from Figure 1. Except for the UM-SCC-11b cells no increase in cisplatin sensitivity was seen for HPV neg. cell lines when pretreated with BEZ235, in contrast to the strong enhancement obtained for all HPV pos. cell lines (Figure 3c). These data demonstrate that the impact of BEZ on cisplatin sensitivity does not depend on its lethal effect, and it is qualitatively different for HPV neg. and pos. cell lines. When normalized to the effect of cisplatin alone, the enhancement caused by BEZ235 was significantly different between the HPV neg. and pos. cells (Figure 3d, *p* = 0.001).

The clear enhancement of cisplatin seen for HPV pos. cells after pretreatment with BEZ235 suggests that this might also result from defective HR. This assumption was tested by suppressing HR in three HPV neg. cells using, again, KD of RAD51. Figure 3e shows, that for two HPV neg. cell lines (Cal-33 and UM-SCC-3) the KD led to a significant enhanced effect of the combined treatment by cisplatin and BEZ235—similar that which has been previously observed for HPV pos. cells. Surprisingly, for UM-SCC-11b, which was the only HPV neg. cell line where the combined treatment sensitized the cells, KD of RAD51 was found to reverse this effect, indicating that this cell line is not defective in HR, but appears to have another DSB repair defect (Figure 3e).

### 3.4. BEZ235 Leads to a Synergistic Effect of Cisplatin and Radiation for Both HPV Neg. and HPV Pos. Cells

It was of special interest to test whether BEZ235 can also be used to enhance the combined effect of cisplatin and radiation. Figure 4a,b shows the data obtained for the HPV neg. cell line Cal-33, as well as the HPV pos. cell line, UD-SCC-2. For cisplatin, the respective IC50 was always used. Note that cell survival was always normalized to the non-irradiated cells. For Cal-33 cells, cisplatin was found to result in a slight radiosensitization, which, however, was significant only at 4 Gy. In contrast, for UD-SCC-2, cisplatin was measured to result in a modest degree of radioresistance, which was already apparent at 2 Gy (Figure 4a,b, open vs. grey circles). For both cell lines, pretreatment with BEZ235 was found to result in a strong radiosensitization, which is in line with previous results [38]. Most surprisingly, when pretreatment with BEZ235 was added to the combination of radiation and cisplatin, a further increase in radiosensitivity was seen for both cell lines (Figure 4a,b, grey and black squares). These data demonstrate that for UD-SCC-2 cells, pretreatment with BEZ235 was even found to convert the negative effect of cisplatin on radiosensitivity into a positive effect by causing clear radiosensitization.

Figure 4c,d show the effect of cisplatin on survival at 2 Gy without (Figure 4c) and with (Figure 4d) pretreatment with BEZ235 for four HPV neg. and five HPV pos. cell lines, respectively. Except for UM- SCC-11b, cisplatin was found to have no significant effect on survival at 2 Gy. Pretreatment by BEZ235 always resulted in a strong further reduction in cell survival at 2 Gy. However, more important, when pretreated with BEZ235, cisplatin was found always to result in a further significant decrease in survival when combined with 2 Gy. These data demonstrate that independent of the HPV status and the cell line used, pretreatment by BEZ235 was always able to convert the additive effect of cisplatin into a clear radiosensitization.

Figure 4e shows the relative effect of survival at 2 Gy achieved by cisplatin, as obtained after normalization to the survival after 2 Gy alone, without or with pretreatment with BEZ235. The plot demonstrates that after pretreatment with BEZ235, the relative effect of cisplatin was even stronger for HPV pos. cells when compared to HPV neg. cells (HPV pos.: *p* = 0.0001 vs. HPV neg. *p* = 0.03).

### 3.5. Radiosensitization Caused by Cisplatin after Pretreatment with BEZ235 Can Be Attributed to Reduced DSB Repair in G1

As demonstrated previously, the increase in cell killing achieved when BEZ235 was added prior to irradiation could be attributed to an enhanced number of residual DSBs, as detected by the co-staining of γH2AX/53BP1 foci [38]. This association, however, was only seen for G1 cells, leading to the conclusion that BEZ235 primarily affects NHEJ, which is the main DSB repair pathway that is active in G1 [48]. The specific effect on NHEJ was confirmed by using a specific DSB repair assay [38].

Figure 5a shows the images of co-stained γH2AX/53BP1 foci obtained for UM-SCC-47 cells after the different treatments used. In non-treated cells, only few foci are present (Figure 5a, first row). After treatment with cisplatin alone, some more 53BP1 foci are visible, but most of all, an almost pan-nuclear signal with phosphorylated γH2AX was apparent in many cells (Figure 5a, 2nd row; hyper-phosphorylation, hp). This signal, which was solely observed for larger nuclei in contrast to small G1 nuclei, is considered to reflect the replication stress in S phase cells resulting from cisplatin-induced DNA damage [49,50]. The percentage of cells with strong γH2AX-phosphorylation (defined as nuclei with >30 γH2AX foci) are depicted in Figure 5b for the HPV pos. cell line UM-SCC-47 and the HPV neg. cell line UM-SCC-3. For both cell lines, these numbers are very low when not treated by cisplatin. After the administration of cisplatin alone, the fraction of nuclei with a hp-signal increased to 47 ± 7% for UM-SCC-3 cells, and even to 69 ± 12% for UM-SCC-47 cells. The fractions slightly decreased when cisplatin was combined with irradiation and/or BEZ235. This is because after combined treatment with BEZ235 or irradiation, some cells will be blocked in G1 or G2, respectively, thereby reducing the fraction of S phase cells [38].

In nuclei with hp-signals, the co-localization of γH2AX and 53BP1 foci was less than 20% (Figure 5d). In contrast, for G1 cells, which were identified by a diameter < 10 µM (see scale bar in Figure 5a), co-localization was seen for almost 90% of the foci. As a consequence, foci counting had to be restricted to G1 cells. The respective numbers are shown in Figure 5c. Irradiation with 2 Gy alone led to a slight increase in the number of foci. There was, however, a strong rise, when cells were pretreated with BEZ235 prior to irradiation. This effect was shown to result from a depressed rate of DSB repair in G1 [38].

Cisplatin alone only resulted in a moderate increase in the number of foci counted in G1 cells (Figure 5c, first two columns of the second block). These numbers slightly increased when cisplatin was combined with irradiation. However, a very strong enhancement was seen when cells were additionally pretreated by BEZ235. The resulting numbers were even higher when compared to the respective numbers obtained without cisplatin, with a significance for the HPV neg. but not for the HPV pos. cell line (Figure 5c). These data suggest that the effect of BEZ235 appears to be even stronger when added to the combined treatment of radiation and cisplatin when compared to the effect on radiation alone.

Figure 5e shows that for both cell lines, the numbers of co-localized foci, as detected for G1 cells after the different treatments, and after correction for the respective background, are well-correlated with the respective cell survival rate taken from Figure 4. That is, for either cell line, an increase in the number of foci detected in G1 always corresponds to a respective decrease in survival. This effect appears to be slightly stronger for the HPV pos. cell line. A similar finding was previously already made for the HPV neg. and pos. HNSCC cells when BEZ235 was given prior to irradiation alone [38]. It is now found here that this strong association between an increased number of residual foci, as measured in G1, and a reduction in cell survival also holds when BEZ235 is added to the combination of both radiation and cisplatin.

## 4. Discussion

Head and neck cancers (HNSCCs), especially in the advanced stage, are treated by radiochemotherapy, including cisplatin, but with a poor outcome for HPV neg. HNSCCs. Moreover, this treatment is often accompanied by severe side effects primarily resulting from cisplatin administration. Therefore, the aim of this study was to test whether BEZ235, which is an effective dual inhibitor of the PI3K/mTOR signaling pathway, can be used to enhance the effect of cisplatin alone, but especially when combined with radiation.

The study was performed with six HPV neg. and six HPV pos. HNSCC cell lines to cover the molecular heterogeneity that is well-known for this entity [41]. The cell lines used were previously shown to reflect the clinical response to both radiation as well as cisplatin [40,46,51,52].

Overall, cisplatin sensitivity was characterized by a huge degree of variation, with HPV pos. cells being slightly more sensitive when compared to HPV neg. cells, albeit without reaching significance. However, when the data was analyzed together with data from other reports using a similar protocol [53,54], an overall trend of a higher cisplatin sensitivity was found for HPV pos. cells. This finding demonstrates, that similar to that found for radiation, a difference in cisplatin sensitivity is seen between HPV pos. and neg. HNSCC cells lines only when a great number of cell lines is analyzed. This reflects the huge heterogeneity of both entities [41,46,52,55]. In contrast, when using viability tests for both radiation as well as cisplatin, a lower degree of sensitivity was measured for HPV pos. cells when compared to HPV neg. cell lines [56,57]. This finding indicates that this test does not measure the actual cellular sensitivity, probably because it also depends on proliferation.

It is shown here for the first time that the higher cisplatin sensitivity of HPV pos. cells might in part result from their defective HR. This defect was recently verified for HPV pos. cells via a lack of formation of RAD51 foci after irradiation [40], which is a key step of HR [58]. Both entities show similar degrees of expression of RAD51, but for HPV pos. cells, the KD of RAD51 was found to have no effect on cisplatin sensitivity, while a clear increase was observed for HPV neg. cells.

HR is involved in the repair of cisplatin-induced DSBs [9,10,11,12] and cells that are defective in HR are characterized by a higher degree of cisplatin sensitivity [59,60,61]. In line with this, for HeLa cells, KD of RAD51 was found to result in an increased degree of cisplatin sensitivity [62], and for NSCLC cells, impaired RAD51 foci formation was associated with enhanced cisplatin sensitivity [59].

Concerning the heterogeneous cisplatin sensitivity, there is great interest in establishing robust markers for prediction [63,64]. So far, only a few markers are known, such as the expression of ERCC1 [65] and the FA/BRCA pathway [66]. It is now shown here for HNSCC that HPV positivity might also be a useful marker.

We also found that cisplatin sensitivity is strongly enhanced in HPV pos. but not HPV neg. cells, when pretreated with BEZ235. This difference in enhancement could likewise be attributed to the defect of HR in HPV pos. cells. When HR was also depressed in HPV neg. cells via KD of RAD51, a similar degree of enhancement of cisplatin by BEZ235 was obtained, as for HPV pos. cells.

It was recently shown by us that at a concentration of 50 nM, BEZ235 NHEJ is suppressed in HNSCC cells, while HR is not affected [38]. NHEJ is also known to be involved in the repair of DSBs resulting from cisplatin-induced intra-strand crosslinks [11,67]. When NHEJ is depressed, more cisplatin-induced DNA damage will collapse with replication, which will require repair via HR. As a consequence, more cell killing is induced in the HR deficient HPV pos. cell lines. In contrast, HPV neg. cells are still able to cope with this additional damage, except that HR is suppressed by KD of RAD51.

The enhancement of cisplatin sensitivity seen for the HPV neg. cell line UM-SCC-11b when pretreated with BEZ235 does not result from a defective HR. It was previously shown that HR is functional in UM-SCC-11b, as demonstrated by an active formation of RAD51 foci after irradiation, and by its response to the CDK inhibitor roscovitine [40]. Surprisingly, in these cells, the enhancement of cisplatin sensitivity by BEZ235 was completely abolished when HR was suppressed via KD of RAD51. These data suggest that UM-SCC-11b may shift to another pathway when HR is depressed, and thereby is able to rescue from the BEZ235 effect. Probably, DSB repair is regulated in these cells due the downregulation of ATM, Mre11, and H2AX as measured for UM-SCC-11b [68].

The most important observation made here is that BEZ235 can be used to enhance the combined effect of cisplatin and radiation in HNSCC cells. In contrast, no such enhancement was seen for HNSCC cells, when cisplatin and radiation were combined with the multi-kinase inhibitor sorafenib [69]. For HNSCC cells, only additive effects were generally seen when cisplatin was combined with radiation, and for some cells, even a decrease in radiosensitivity was measured [53,54,70]. The strong effect of this combination seen in clinical studies [5,6,71] might result from the complementary action of these two, with cisplatin being especially effective in proliferating cells [13,72], which, on the other hand, are known to be rather radioresistant [14].

The enhanced cell killing effect of the combined treatment with cisplatin and radiation when using a pretreatment with BEZ235 was seen for both HPV neg. and pos. cells, with a slightly stronger effect for the latter. This enhancement is considered partly to result from a diminished degree of DSB repair in G1, as demonstrated for two cell lines by the correlation between the decrease in survival and the increase in residual DSBs, as detected in G1 cells by co-staining with γH2AX and 53BP1. Since BEZ235 inhibits NHEJ and also induces a G1-arrest, as shown previously [38], more cisplatin-induced DSBs will remain unrepaired in G1 when cells are pretreated with BEZ235. Consequently, there is a higher chance for a lethal interaction, with radiation-induced damage leading to a higher level of residual DSBs, and with that, to more cell killing, as observed when cells are pretreated with BEZ235 prior to the combination of cisplatin and radiation. However, some of the unrepaired cisplatin-radiation adducts will also interfere with replication, therefore causing more cell killing in HR-deficient HPV pos. cells. This explains why the enhancement of the combined treatment with radiation and cisplatin by BEZ235 was stronger for HPV pos. cells.

## 5. Conclusions

Overall, our data indicate that BEZ235 might be used as an excellent tool to enhance the combined effect of cisplatin and radiation. This enhancement might even allow for a reduction in the amount of cisplatin given to the patient, thereby reducing the side effects, but still reaching an excellent response. Unfortunately, BEZ235 has slipped from focus because of the negative results obtained when it is given as monotherapy (NCT01856101). This is a big drawback, because other PI3K inhibitors such as BKM120 and GDC0980 appear to be less efficient [37]. For the combination of BKM120 and RT, a clinical study with HNSCC was just completed, and the respective results will be expected in the near future (NCT02113878). The results presented here might help to initiate new interest in the potent inhibitor BEZ235, in order to enhance the combined effect of cisplatin and radiation when starting a new clinical trial with HNSCC.

## Figures and Tables

**Figure 1 cancers-14-03160-f001:**
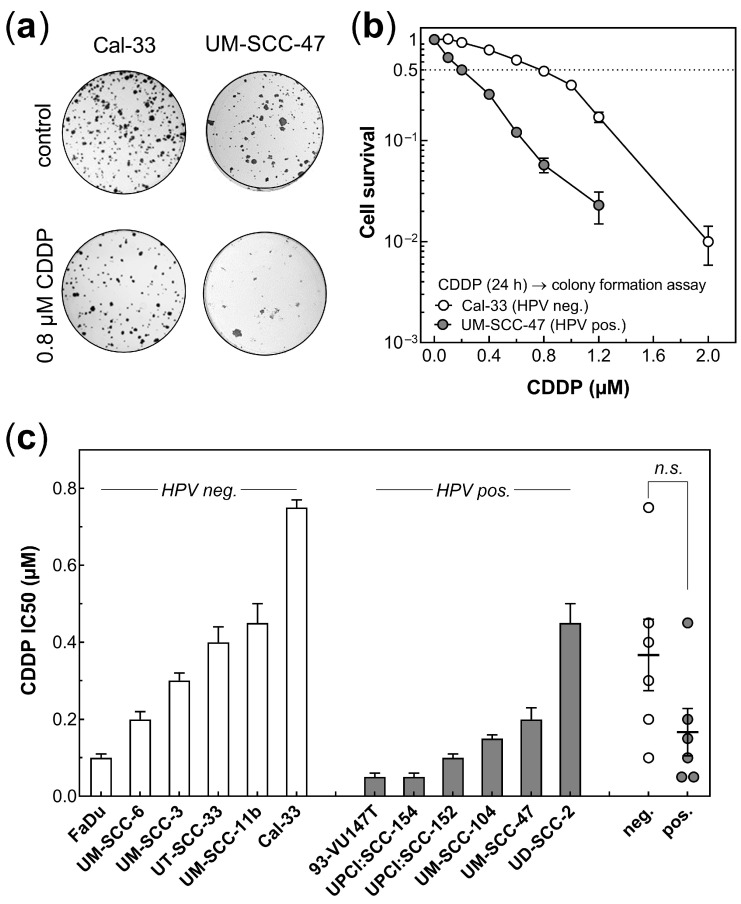
Cisplatin sensitivity of HPV neg. and HPV pos. HNSCC cell lines. (**a**) Cells were treated with cisplatin (CDDP) for 24 h, with concentrations of up to 2.0 µM, and survival was measured by a colony formation assay. (**b**) Cell survival for Cal-33 and UM-SCC-47 cells. (**c**) IC50 determined for cisplatin for six HPV pos. and six HPV neg. HNSCC cell lines using the respective survival curves. A Mann–Whitney test was used for statistical analysis. Data are presented as mean values ± SEM, n ≥ 3, *n.s.*: non significant.

**Figure 2 cancers-14-03160-f002:**
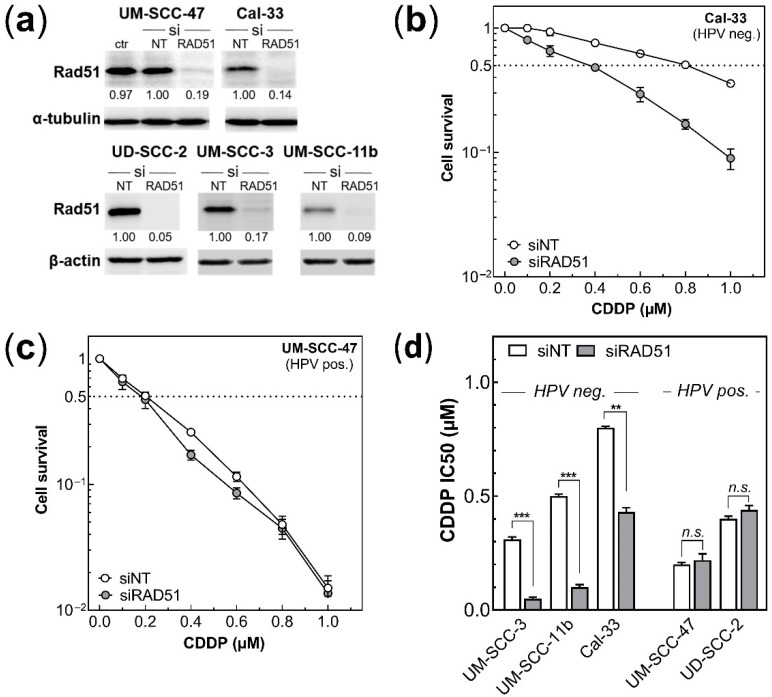
Effect of RAD51 KD on cisplatin sensitivity in HPV neg. and HPV pos. HNSCC cell lines. (**a**) RAD51 level 24 h after transfection with siRAD51 or siNT (non-target control) or native (ctr) cells; α-tubulin and β-actin were used as loading controls. (**b**,**c**) Cisplatin sensitivity of Cal-33 and UM-SCC-47 cells, with or without the knock-down of RAD51. Cells were treated with cisplatin (CDDP), as described in Figure 1. (**d**) Impact of RAD51 knock-down on IC50 of cisplatin, as determined for three HPV neg. and two HPV pos. HNSCC cell lines. Data are presented as mean values ± SEM, n ≥ 3, ** *p =* 0.001; *** *p* = 0.0001, *n.s.*: non significant.

**Figure 3 cancers-14-03160-f003:**
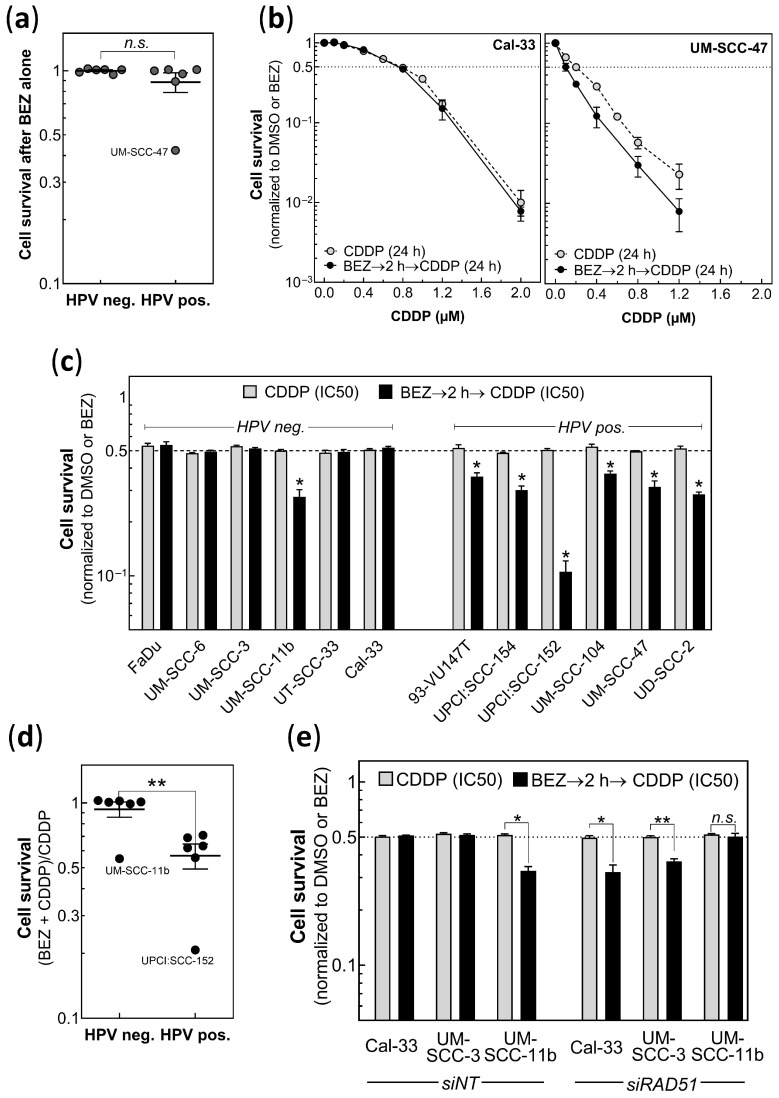
Effect of BEZ235 on the cisplatin sensitivity of both HPV neg. and HPV pos. HNSCC. Cells were pretreated with 50 nM BEZ235 (BEZ) for 2 h before incubation with cisplatin (CDDP) for 24 h. (**a**) Effect of BEZ235 alone. (**b**) Effect of pretreatment with BEZ on the cisplatin sensitivity of Cal-33 and UM-SCC-47 cells, respectively. (**c**) Effect of cisplatin alone or after pretreatment with BEZ235, as determined for all six HPV neg. and pos. cell lines. For cisplatin, the respective IC50 doses were used as taken from Figure 1. (**d**) Relative reduction in cell survival as achieved for HPV neg. or pos. cell lines by the combined treatment of BEZ235 and CDDP after normalization to the effect of cisplatin alone. (**e**) Effect of RAD51 KD (siRAD51) compared to non-target control (siNT) on the combination of BEZ235 and cisplatin for HPV neg. HNSCC cell lines using the IC50 concentrations as shown in Figure 2d. (**b**,**c**,**e**) Cell survival is always normalized to cells non-treated by CDDP. Data presented as mean values ± SEM, n ≥ 3, * *p* = 0.05; ** *p* = 0.001, *n.s.*: non significant.

**Figure 4 cancers-14-03160-f004:**
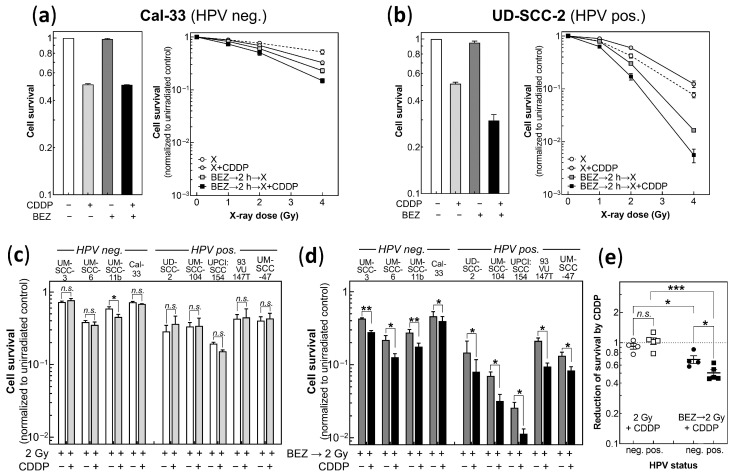
Effect of BEZ235 on the combined treatment of HPV neg. and HPV pos. HNSCC cell lines with cisplatin and radiation. Cells were pretreated with 50 nM BEZ235 (BEZ) for 2 h before irradiation (X) and incubation with cisplatin (CDDP) using the respective IC50, followed by a medium change after 24 h for colony formation. (**a**) Effect on the HPV neg. cell line Cal-33, and (**b**) on the HPV pos. cell line UD-SCC-2. (**c**) Effect of cisplatin on cell survival at 2 Gy without and (**d**) with pretreatment by BEZ235. Survival was normalized to the non-irradiated cells. (**e**) Relative effect of cisplatin on survival at 2 Gy obtained by normalizing to respective survival after 2 Gy alone, without or with pretreatment with BEZ235. Data are presented as mean values ± SEM, n ≥ 3, * *p* = 0.05; ** *p* = 0.001, *** *p* = 0.0001, *n.s.*: non significant.

**Figure 5 cancers-14-03160-f005:**
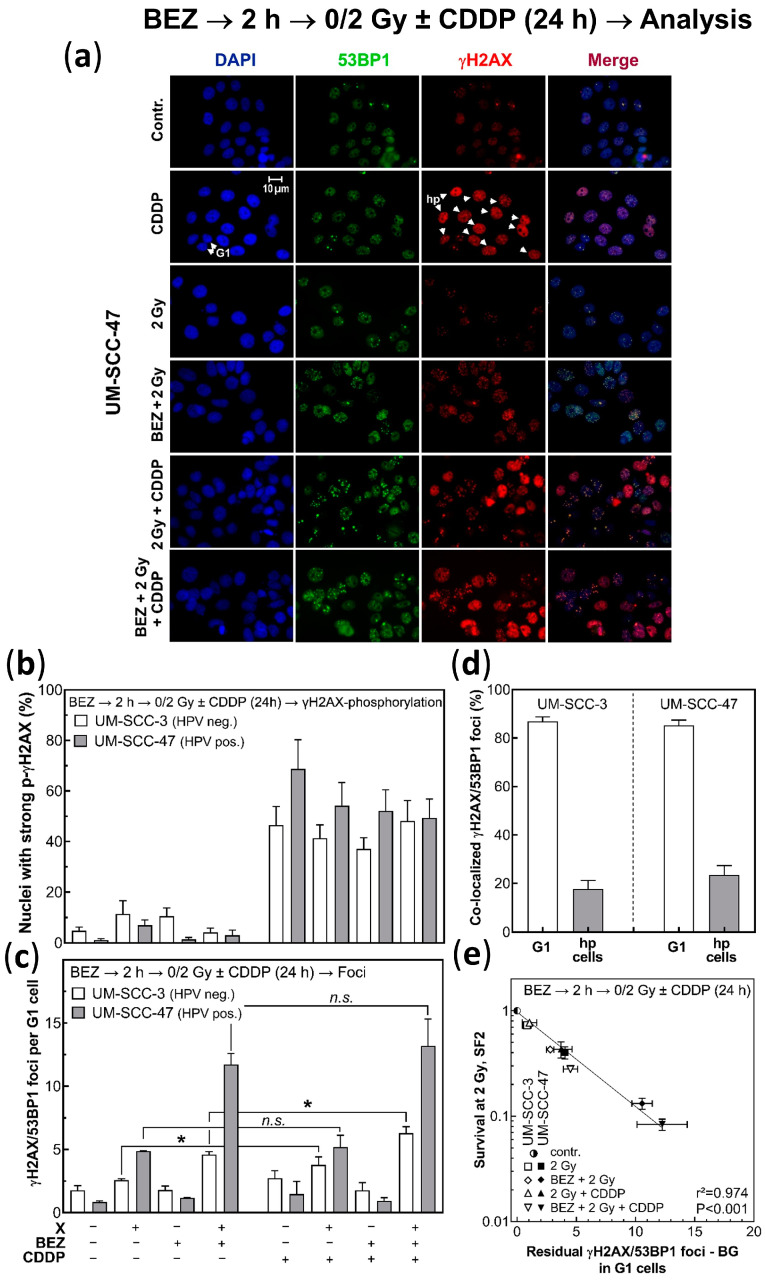
Effect of BEZ235 and the combined treatment with cisplatin (CDDP), and irradiation (X) on γH2AX/53BP1 foci in UM-SCC-3 and UM-SCC-47 cells. Cells were treated with/without 50 nM BEZ235 (BEZ) 2 h before irradiation with 2 Gy and cisplatin (CDDP, IC50) treatment. (**a**) Images as taken for UM-SCC-47 cells; G1 cells, identified by a small nuclear size; hp, hyper-phosphorylation of γH2AX with >30 foci. Enlarged images are presented in Appendix A. (**b**) Percentage of nuclei with hp-γH2AX. (**c**) Number of residual γH2AX/53BP1 foci counted for G1 cells, as identified via a diameter of less than 10 µM (see scale bar in Figure 5a). (**d**) Percentage of co-localized γH2AX/53BP1 foci as measured for G1 cells and hp nuclei, respectively. (**e**) Association between residual number of γH2AX/53BP1 foci after background subtraction and respective cell survival, taken from Figure 4. Data were analyzed by a log-linear regression analysis. Data are presented as mean values ± SEM, n ≥ 2, per value at least 100 cells are counted; * *p* = 0.05; *n.s.*: non significant.

## Data Availability

Not applicable.

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
