# Peer review of "Dual PI3K/mTOR Inhibitor NVP-BEZ235 Leads to a Synergistic Enhancement of Cisplatin and Radiation in Both HPV-Negative and -Positive HNSCC Cell Lines"

_cancers, 2022, doi:10.3390/cancers14133160_

Round 1

Reviewer 1 Report

The authors have sufficiently clarified my questions. Still a well-conducted study that reads well and has a stringent red line. The results are convincing. Since my knowledge of cell culture is limited, I strongly recommend additional review by a cell culture expert to verify this aspect of the work.

Author Response

We like to thank Reviewer 1 for his valuable comments.

Reviewer 2 Report

The resubmission by Subtil et al. is on the PI3K/mTOR inhibitor NVP-BEZ235 in combination with cisplatin and radiation in both HPV negative and HPV positive HNSCC cell lines. The authors have addressed the initial concerns and have dramatically improved the clarity of the results presented. Adding supplementary data, proper control data, increasing the font in the graphs, and the addition of better defined y-axis labels have made the results much easier to interpret. As now presented, it is much easier to deduce the synergistic effects.

Remaining concerns:

In regards to figure 5 and the corresponding section 3.6.

Section 3.6 is hard to read as written. It would be useful to rework this section (a few examples are provided in minor comments below).

The authors make their G1 assumption based on cell diameter. Are there references that can be provided for this way of assigning cell cycle phases? If not, are there other methods to assure these cells are in G1 that are being counted? For example, there are numerous “larger” cells in BEZ+2Gy and 2Gy+CDDP however they also appear to have colocalized foci. In lines 327-330, it is stated that only 20% of cells not in G1 have these foci. These data do not seem to correspond.

It is also worth showing all the images for all the conditions summarized in 5c – the current figure 5a does not show all 3 drugs combined – and this is the major drug combo for your entire study.

Minor comments are below.

Figure 3 legend: the subsections are incorrectly labeled – missing (d) in line 261 and throwing off the subsequent sections. Just change the current “(e) and (f)” to (d) and (e).

Line 315 – check order of letters (spelling of) γH2AX

Line 322 – “After administration of cisplatin alone” there should be a comma after this before “fraction”.

Lines 335 refers to columns 9 and 10 in the graph but they are not numbered on the graph. It might be worth labeling the column numbers on the graph or have another way of delineating these.

Lines 373-378, this sentence is rather confusing and worth breaking up. Or if there is a period after “HPV pos. cells. – it is confusing to start a sentence with I.e.

Reviewer 3 Report

Dear authors,

Thank you for taking into account most of my previous comments. You will find below my last minors points :

* In response to my question regarding the actual number of cells you counted in your DNA damage quantification, I appreciate your fixed the issue. However, the number you reported (100 cells) has to be justified. This could be done via the Terrific app  (https://radbiolab.shinyapps.io/terrific/ ; doi.org/10.1667/RADE-20-00165.1 ) as mentioned previously. The number of 100 cells is fully compatible with your experimental design (triplicate, 24h post-irradiation, 100% confluence and delivered dose > 0.5 Gy).

* I do not find any answer to my comment regarding residual y-H2AX foci. I copy/paste my previous comment on this point :In the discussion, you seem to associate residual y-H2AX foci with residual DSBs. According to the guidelines for quantification of DNA damage (doi.org/10.1093/narcan/zcab046), this is increasingly criticized in the literature. This statement should therefore be nuanced.

Author Response

This manuscript is a resubmission of an earlier submission. The following is a list of the peer review reports and author responses from that submission.

Round 1

Reviewer 1 Report

First, I would like to thank the authors for allowing me to review your manuscript "Dual PI3K/mTOR Inhibitor NVP-BEZ235 Leads to a Synergistic Enhancement of Cisplatin and Radiation in both HPV Negative and Positive HNSCC Cell Lines".

In the in vitro cell line study, Subtil et al. used 12 different HNSCC cell lines (6 HPV+ and 6 HPV-) to investigate the dual inhibitor NVP-BEZ235 (BEZ235, dactolisib). This can obtain a strong suppression of the dual PI3K/mTOR pathway.

Overall, a well-conducted study that reads well and has a stringent red line. The results are convincing. Since my knowledge regarding cell culture is limited, I recommend an additional review by an expert in cell culture to review this aspect of the work.

Major points:

- In Figure 1 c, a t-test? was done that was not significant. I am not sure, I rather think a Mann-Whitney-U test is appropriate. I do not believe that the data are normally distributed. In Figure 1, 2, 3, 4 the mean values +/- SEM is given each time. However, this can be deceptive about the true distribution of the data. The authors should use +/- SD. Boxplots are even better to show the data distribution.

- A p-adjustment should be performed (Bonferroni, Holm, etc.).

Minor points:

- The authors write in the First paragraph of the Introduction that HNSCC for HPV positive cases has a cure rate of 85%. This may be true for the oropharynx, but not for the larynx or oral cavity. The authors need to be precise here and correct it. 

- Figure 5 is too small, as are other figures. You can hardly read the text. Please make it bigger. Also (d) is positioned strangely. I would put (a) on top, and (b), (c), (d) next to each other, but bigger and using the whole page width. So please make all figures bigger.

Reviewer 2 Report

This work by Subtil et al. is on the PI3K/mTOR inhibitor NVP-BEZ235 in combination with cisplatin and radiation in both HPV negative and HPV positive HNSCC cell lines. Standard of care treatment for HNSCC has severe side effects, so it is important to find adjuvants that will synergize with current treatment paradigms to lessen the dose needed for good patient outcomes. Collectively this data suggests that BEZ-235 may provide such an opportunity. However, the data as presented is underdeveloped, does not show proper controls, and lacks proper synergistic style assays to confirm the conclusion/title that “BEZ235 leads to a synergistic enhancement of cisplatin and radiation in both HPV negative and Positive HNSCC”. Moreover, the background and citations are lacking for the field of HPV related HNSCC – which respond entirely different to treatments in the clinic. Concerns are outlined below.

Major concerns

  1. Figure 1 – While Figure 1C summarizes the CDDP IC50 of cisplatin, it would be beneficial to the reader to see every cell line in the same way as shown with Cal-33 and UMSCC47 in Figure 1A and 1B, at bare minimum as supplemental data. The authors are also inconsistent with the cell lines utilized as “representative” images throughout this manuscript.
  2. Supplemental figure S1A is an excellent addition to show the baseline levels of Rad51 in all cell lines. Is there a reason why 3 HPV negative and 2 HPV positive cell lines were chosen for Rad51 knockdown in Figure 2? Were the cells chosen at random, were they the cells with the best knockdown? The authors are unclear why these cells were selected over the other cells utilized in other figures in this paper. For example, SCC152 and SCC154 are more sensitive in Figures 3 and 4 but were not chosen for Rad51 knockdown.
  3. Figure 3 – While BEZ alone is shown in Figure 3C, it would be much more beneficial to be shown in the same graphs as the other treatments – same critique for Figures 4 and 5 BEZ alone should be shown, this data is completely missing. This is an important thing for readers to see each drug alone in these figures, even if previously published in other reports. There is also one cell line that seems to be particularly sensitive to BEZ alone in Figure 3C, but it is never discussed. Is there a reason why SCC152 cells in Figure 3D are more sensitive than others in this combinational approach? This is never discussed.
  4. The authors state that BEZ235 sensitization can be contributed to reduced DSB repair in G1. They show γH2AX/53BP1 foci as evidence of DSB in Figure 5, however there is minimal evidence that this is occurring in G1 other than citing a previous publication that did not use the drug+ radiation combinational approach in this manuscript. Radiation can alter cell cycle and this needs to be monitored via FACS like the previous publication. The images are also too small to clearly see the effects the authors are reporting.
  5. The authors state that these combinational approaches lead to a synergistic response. However, the data as presented is not analyzed in a way that proves synergism. Tallarida RJ. Quantitative Methods for Assessing Drug Synergism. 2011. https://www.ncbi.nlm.nih.gov/pmc/articles/PMC3379564/ is a methods paper that can be referred to for these proper calculations.

Minor Concerns

  1. Both the simple summary, abstract, and introduction utilize essentially the same sentence: “This treatment results in a 5-year-survival rate of about 85% for HPV-positive HNSCC but only 50% for HPV negative HNSCC”. It is important that the authors not use the same sentence over and over.
  2. In the introduction near line 59 – it is worth mentioning that HPV promotes the maintenance of S-phase in cells (for viral replication), this is a likely mechanism by which Cisplatin sensitivity is present in HPV positive HNSCC.
  3. Methods section 2.1 lines 95-107. UMSCC104 cells need to be grown in EMEM media. Was this media left out in the methods? It is our experience that there are numerous issues with their culture if they are not grown in their proper media.
  4. Line 191. States that HPV positive cell lines are on average slightly more sensitive than HPV negative cells – however the data is not significant and should be stated that while there is a trend for sensitivity, it is not significant as presented (Figure 1C).
  5. The figure labels throughout are often too small of font to be clearly legible for the reader. It would be beneficial to increase the font size in many of the graphs. A specific example would be in Figure 3A, the exponential numbers on the y-axis are very hard to see clearly, and Figure 4E, it is hard to determine if the graph label says n.s. or if there is significance.

Reviewer 3 Report

In their manuscript entitled "Dual PI3K/mTor inhibitor NVP-BEZ235 leads to a synergistic enhancement of cisplatin and radiation in both HPV negative and positive HNSCC cell lines", authors showed interesting data regarding the differential impact of BEZ235 on cisplatin efficiency in HPV positive & negative cells. Interestingly, they showed that pre-incubation with BEZ235 modify the usual additive effect observed when cisplatin is combined with radiotherapy.
Please find below my comments :

* Supplemental figure 1 : Could you add a table with the value of decreased Rad51 expression following knocked-down in each cell type (proportion regarding basal expression)?

* I suggest that some of the intermediate conclusions in the results be nuanced because at this point in the reading, the reader does not have all the information needed to make these statements as clearly. Example (line 208-211) : "KD of Rad51 ... demonstrated that the higher cisplatin sensitivity of HPV positive cells was in fact due to their defect in HR".
A nuance like "seems to be linked to" seems to me more appropriate.

* Figure 2A : I assume that values written on the western blot result from a quantification. Why do you use actine for some samples and tubulin for other ones ?

* Section 3.6. : How do you define (on which basis ?) a "strong" y-H2AX phosphorylation or a "small" nuclear size ? I do not find reference to this in the Mat & Met section.

* Figure 5A : Could you provide scale bar on images ?

* In the legend of figure 5, you mentioned that at least 50 cells were counted (line 347). However in Mat & Met section, you mentioned at least 100 cells (lines 175). I'm a little confused about the actual number of cells you counted. Can you solve this ?
When I calculate the number of cells you have to count to ensure sufficient statistics via the Terrific app ( https://radbiolab.shinyapps.io/terrific/ ; doi.org/10.1667/RADE-20-00165.1 ), I find the value of at least 70 cells for your experimental design (triplicate, 24h post-irradiation, 100% confluence and delivered dose > 0.5 Gy). A reference to this tool in the Mat and Met section would help to clarify your choice of the number of cells to count. 

* In the discussion, you seem to associate residual y-H2AX foci with residual DSBs. According to the guidelines for quantification of DNA damage (doi.org/10.1093/narcan/zcab046), this is increasingly criticized in the literature. This statement should therefore be nuanced. 

* I assume that you have chosen the colocalization between y-H2AX and 53BP1 as the metric for your study because, according to the aforementioned guidelines, it is the most accurate metric to date. Nevertheless, I would be interested in having the proportion of yHAX-positive/53BP1-positive foci and the proportion of yH2AX-positive/53BP1-negative foci, which would be a metric of the proportion of damage repaired by NHEJ versus other pathways, respectively. 

Congratulation for this nice work !